# Spatio-Temporal Variation Analysis of the Biological Boundary Temperature Index Based on Accumulated Temperature: A Case Study of the Yangtze River Basin

**Guangxun Shi** [1,2,3,4], **Peng Ye** [5,6,*] and **Xianwu Yang** [7]

1    School of Geography, Nanjing Normal University, Nanjing 210023, China; 161301021@njnu.edu.cn
2    Key Laboratory of Virtual Geographic Environment, Ministry of Education, Nanjing Normal University, Nanjing 210023, China
3    Jiangsu Center for Collaborative Innovation in Geographical Information Resource Development and Application, Nanjing 210023, China
4    State Key Laboratory Cultivation Base of Geographical Environment Evolution (Jiangsu Province), Nanjing 210023, China
5    Urban Planning and Development Institute, Yangzhou University, Yangzhou 225127, China
6    College of Civil Science and Engineering, Yangzhou University, Yangzhou 225127, China
7    School of Geographic Sciences, Xinyang Normal University, Xinyang 464000, China; yangxianwu@xynu.edu.cn
*    Correspondence: 007839@yzu.edu.cn; Tel.: +86-156-5195-8693

**Abstract:** Active accumulated temperature is an important index of agricultural heat resources in a region. Based on the temperature data of the Yangtze River Basin from 1970 to 2014, this paper analyzed the characteristics of the temporal and spatial variations of the biological boundary temperature in the Yangtze River Basin. The main conclusions were drawn as follows: (1) since 1970, the accumulated temperature of $\geq 0$ °C in the northern subtropical zone, mid-subtropical zone, and plateau climate zone showed overall increasing trends, and the trends were 122 ($p < 0.001$), 87.7 ($p < 0.001$), and 75.3 °C/10a ($p < 0.001$), respectively. The accumulated temperature of $\geq 5$ °C showed an upward trend, and the change tendency rates were 122.6 ($p < 0.001$), 90.5 ($p < 0.001$), and 81.4 °C/10a ($p < 0.001$), respectively. The accumulated temperature of $\geq 10$ °C showed overall increasing trends and the trends were 115.7 ($p < 0.001$), 92.5 ($p < 0.001$), and 78.9 °C/10a ($p < 0.001$). Accumulated temperatures of $\geq 0$ °C, $\geq 5$ °C, and $\geq 10$ °C in the northern subtropical zone increased significantly higher than that in the mid-subtropical zone and plateau climate zone. (2) The accumulated temperatures of $\geq 0$ °C, $\geq 5$ °C, and $\geq 10$ °C in the northern subtropical zone showed an abrupt change in 1997. In the mid-subtropical zone and plateau climate zone, there was an abrupt change in the accumulated temperatures of $\geq 0$ °C and $\geq 5$ °C in 1994, and in the northern subtropical zone, the abrupt change of the accumulated temperature $\geq 10$ °C occurred in 1998. (3) There are obvious differences in the biological boundary temperature within the Yangtze River Basin, and the stations with large increases are mainly distributed in the middle and lower reaches, such as the Hanshui Basin, the Poyang Lake Basin, the Taihu Lake Basin, and the middle and lower reaches of the mainstream area. The initial day, final day, and continuous days showed a trend of advancement, postponement, and extension, respectively. Besides, the heat resources showed significant increasing trends, which is of guiding significance for the future production and development of agriculture in the region. With the increase of heat resources in the Yangtze River Basin, appropriate late-maturing varieties should be selected in variety breeding, to make full use of heat resources and improve the quality of agricultural products. Secondly, the planting system should be adjusted and the multiple cropping index improved to steadily increase agricultural output. This brings new opportunities to adjust the structure of the agricultural industry and increase farmers' income, in the Yangtze River basin.

**Keywords:** accumulated temperature; biological boundary temperature index; climate spatio-temporal change; Yangtze River Basin

## 1. Introduction

Agricultural production is a system highly dependent on natural conditions, especially climate conditions, and global warming inevitably has a profound impact on it [1] Currently, the degree and scope of the impact of global warming on agricultural production has become a focus of research and which is explored by scholars worldwide. The biological boundary temperature plays an important guiding role in phenological phenomena and in agricultural production [2]. Research on active accumulated temperatures of $\geq 0$, $\geq 5$, and $\geq 10\,°C$ and its continuous days are helpful to study the possible growing period of crops in a certain area and have a profound impact on the production of agriculture and animal husbandry [3]. In addition, temperature indicators, such as the initial day, the final day, the continuous days, and effective accumulated temperatures $\geq 0$, $\geq 5$, and $\geq 10\,°C$ are of great significance in discussing the driving mechanism of phenological changes [4]. A temperature of $\geq 0\,°C$ is when cool-season crops start to grow [5]; a temperature of $\geq 5\,°C$ is when high-altitude vegetation turns green and other crops, such as rapeseed, start to grow [6]; and a temperature of $10\,°C$ is the initial temperature at which the main cool-season crops develop [7]. An effective accumulated temperature of $\geq 10\,°C$ and its initial day, final day, and continuous days are important indicators of agricultural heat for guiding agricultural production, which not only reflect the length of the growth period for the main thermophilic plants [8], but also can be used to measure the number of heat sources in a region and has important practical significance in terms of agriculture climate zonation, reasonable allocation of crops, forecasting phenological periods, and pest occurrence periods [9].

Relevant studies have shown that, with a significant increase in temperature, the temporal and spatial distribution of accumulated temperature at $\geq 0$, $\geq 5$, and $\geq 10\,°C$ have undergone obvious changes in some areas of China [10]. Most areas of the Yangtze River Basin have a subtropical monsoon climate, with a warm and humid climate and the same period of rain and heat. The Taihu Plain, Chengdu Plain, Jianghan Plain, Chaohu Region, Dongting Lake Region, and Poyang Lake Region in the basin are all-important commodity grain bases in China, which are important for ensuring Chinese food security. At present, many studies have shown that the temperature in the Yangtze River Basin has increased significantly in recent decades, but these studies mainly focus on changes in average temperature. No reports on the temporal and spatial changes of accumulated temperature have been made, and only some scholars have conducted research based on administrative units [11].

## 2. Materials and Method

### 2.1. Study Area

The Yangtze River Basin (90°33′–122°25′ E, 24°30′–35°45′ N) (Figure 1), including a vast area the containing main stream and tributaries of the Yangtze River, is the third largest basin in the world, with a total area of 1.8 million square kilometers, accounting for 18.8% of China. The Yangtze River Basin spans 19 provincial administrative units of China's three major economic zones (the eastern economic zone, central economic zone, and western economic zone). In addition, the Yangtze River Basin is also an important commodity grain base in China. Overall, the annual average temperature in the Yangtze River Basin is high in the east, low in the west, high in the south, and low in the north. Except for some high-altitude areas, such as the Western Sichuan Plateau and the source of the Yangtze River, most areas of the basin have a subtropical monsoon climate. The northern subtropical zone is located at 28°–33° N, including the Hanshui River Basin, the middle and lower reaches of the Yangtze River, and the Taihu Lake Basin, accounting for about 5.4% of the national land area. The middle subtropical zone is located in the Han River Basin, the middle and lower reaches of the Yangtze River, the south of the Taihu Lake Basin, and the east of the Qinghai Tibet Plateau, accounting for about 16.5% of China's land area. Qinghai Tibet Plateau is mainly located in the plateau area west of 103° E. The division is mainly based on Zheng's research results.

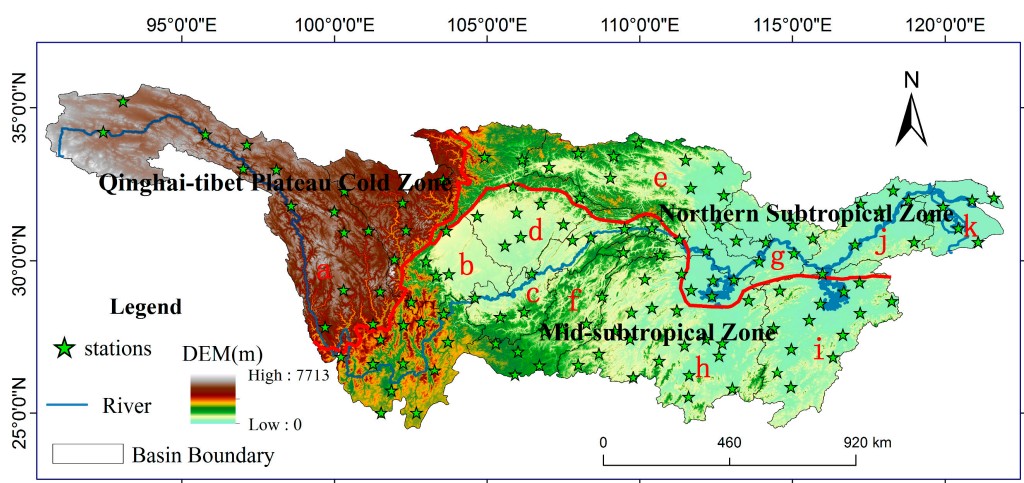

**Figure 1.** Topographic features and meteorological station distribution in the study area. a: Jinsha River Basin, b: Min-Tuo River Basin, c: Upper Main Stream, d: Jiangling River Basin, e: Hanjiang River Basin, f: WuJiang River Basin, g: Midle Main Stream, h: Dongting Lake Basin, i: Poyang Lake Basin, j: Lower Main Stream, k: Taihu Lake Basin.

### 2.2. Data

Considering the variations in the start and end dates of records in the basin's stations and that some stations in the area have been demolished or relocated, this study selected 131 stations in the Yangtze River Basin (Figure 1) to ensure the integrity and consistency of meteorological data and to study the daily average temperature data measured from 1970 to 2014.

The data from the meteorological stations in this study were from the China Meteorological Data Service Centre (http://data.cma.cn/en (accessed on 7 July 2020)), and the daily average temperature was used for the analysis. For the stations without this index, or few data from meteorological stations without measured or with missing climatic elements, a regression method was used for interpolation based on the neighboring stations [12]. The vector data for each climate zone within the study area the northern subtropical zone, the mid-subtropical zone, and the Qinghai-Tibet plateau climate zone were extracted from Zheng's research results [13]. Basin boundary data is mainly extracted based on DEM data with a resolution of 1 km [14,15].

### 2.3. Study Method

#### 2.3.1. Definition of Accumulated Temperature

The active accumulated temperature is an important basis to evaluate the biological boundary temperature index. The method proposed in this paper utilized the currently widely used 5-day sliding average approach to calculate the initial day and the final day of each year, for which the daily average temperature has stably passed $\geq 0$, $\geq 5$, and $\geq 10$ °C. The growing season (continuous days) is determined as the sum of the continuous days between the start of the growing season (initial day) and the end of the growing season (final day), and the active accumulated temperature was determined as the sum of the average temperature between the initial day and the final day [16].

AT0, AT5, and AT10 is defined as the sum of daily mean temperatures above 0, 5, and 10 °C in a continuous period of 1 year. If the temperature values exceed 0, 5, and 10 °C for five consecutive days, then the AT0, AT5, and AT10 is calculated from the first day of those five days and extended until the final day above 0, 5, and 10 °C. Accumulated temperature (AT) was calculated using the following formula:

$$\text{AT } 0,5,10 = \sum_{i=1}^{n} \overline{T}_i \left( \overline{T}_i > 0,5,10 \right) \tag{1}$$

where $T_i$ is the mean air temperature on day $i$; $i = 1, 2, 3, \ldots, n$ and $n$ is the total number of the days when daily mean temperature values were larger than 0, 5, and 10 °C continuously; therefore, $T_i > 0$ °C, $T_i > 5$ °C, and $T_i > 10$ °C.

### 2.3.2. Cumulative Anomaly

Cumulative anomaly is a common method used to judge the change trend directly from the curve. A cumulative anomaly curve showing an upward trend indicate that the anomaly value increases, and a downward trend indicates that the anomaly value decreases. From the obvious ups and downs of the curve, we can judge the long-term trend in evolution and continuous change, and can even diagnose the approximate time of abrupt change.

For sequence x, the cumulative anomaly at a certain time t is expressed as follows:

$$\hat{x}_t = \sum_{i=1}^{t}(x_i - \overline{x}), t = 1, 2, \ldots, n \tag{2}$$

where:

$$\overline{x} = \frac{1}{n}\sum_{i=1}^{n} x_i \tag{3}$$

After calculating all of the cumulative anomaly values at n times, the cumulative anomaly curve was drawn for trend and catastrophe analysis.

## 3. Results and Analysis

### 3.1. Temporal and Spatial Characteristics of Accumulated Temperature $\geq 0$ °C

3.1.1. Temporal Characteristics

(1)   Interannual variation of accumulated temperature $\geq 0$ °C

Figure 2 indicates the interannual variation characteristics of accumulated temperature of $\geq 0$ °C on a regional scale, namely in the northern subtropical zone, mid-subtropical zone, and plateau climate zone from 1970 to 2014. The perennial average accumulated temperatures of $\geq 0$ °C in the three climatic regions were 5712, 5936.9 and 2311.2 °C. From the perspective of the trend for interannual change, since 1970, the accumulated temperature at $\geq 10$ °C in the three climatic regions showed an overall increasing trend, with a change tendency rate of 122 ($p < 0.001$), 87.7($p < 0.001$), and 75.3 °C/10a ($p < 0.001$). The increase in the northern subtropical zone is significantly higher than that in the mid-subtropical zone and plateau climate zone. The accumulated temperature in the northern subtropical zone and plateau climate zone were different for two different periods, which were 1970–1993 and 1994–2014, and the accumulated temperature in the mid-subtropics were different for two different periods, which were 1970–1997 and 1998–2014 (Figure 3).

Before the mid-1990s, the accumulated temperature in the three climatic regions increased by 31.3 ($p = 0.29$), 24.8 ($p = 0.13$), and 18.2 °C/10a ($p = 0.24$), while after the mid-1990s, the accumulated temperature increased at rates of 31.3 ($p = 0.29$), 24.8 ($p = 0.13$), and 18.2 °C/10a ($p = 0.24$). Moreover, the accumulated temperatures increased sharply. Figure 3 shows the accumulated temperature anomalies and accumulated anomalies in the northern subtropical zone, mid-subtropical zone, and plateau climate zone. The results show that the average accumulated temperatures in the northern subtropical zone and plateau climate zone from 1970 to 1993 were negative anomalies. After 1993, the average accumulated temperature was dominated by positive anomalies. On the regional scale, the accumulated temperature in the northern subtropical zone and that of $\geq 0$ °C in the plateau climate zone had abrupt changes around 1993.

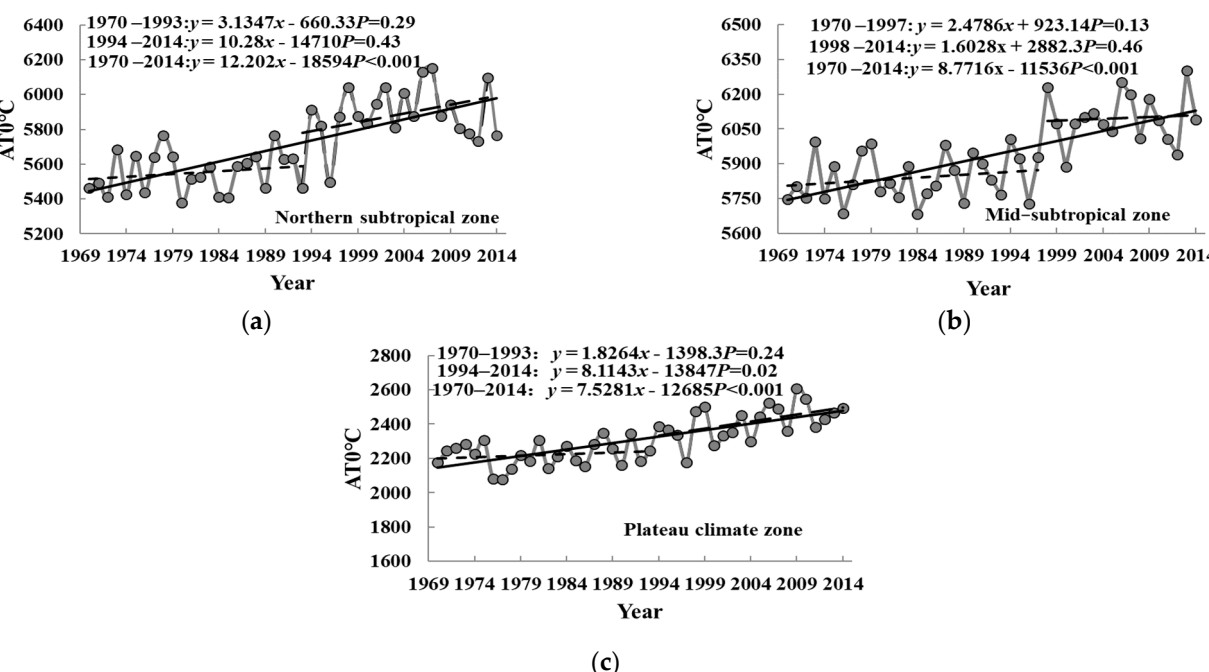

**Figure 2.** Interannual variation in the accumulated temperature at ≥0 °C in the northern subtropical zone (**a**), the mid-subtropical zone (**b**), and the plateau climate zone (**c**) of the Yangtze River Basin during 1970–2014.

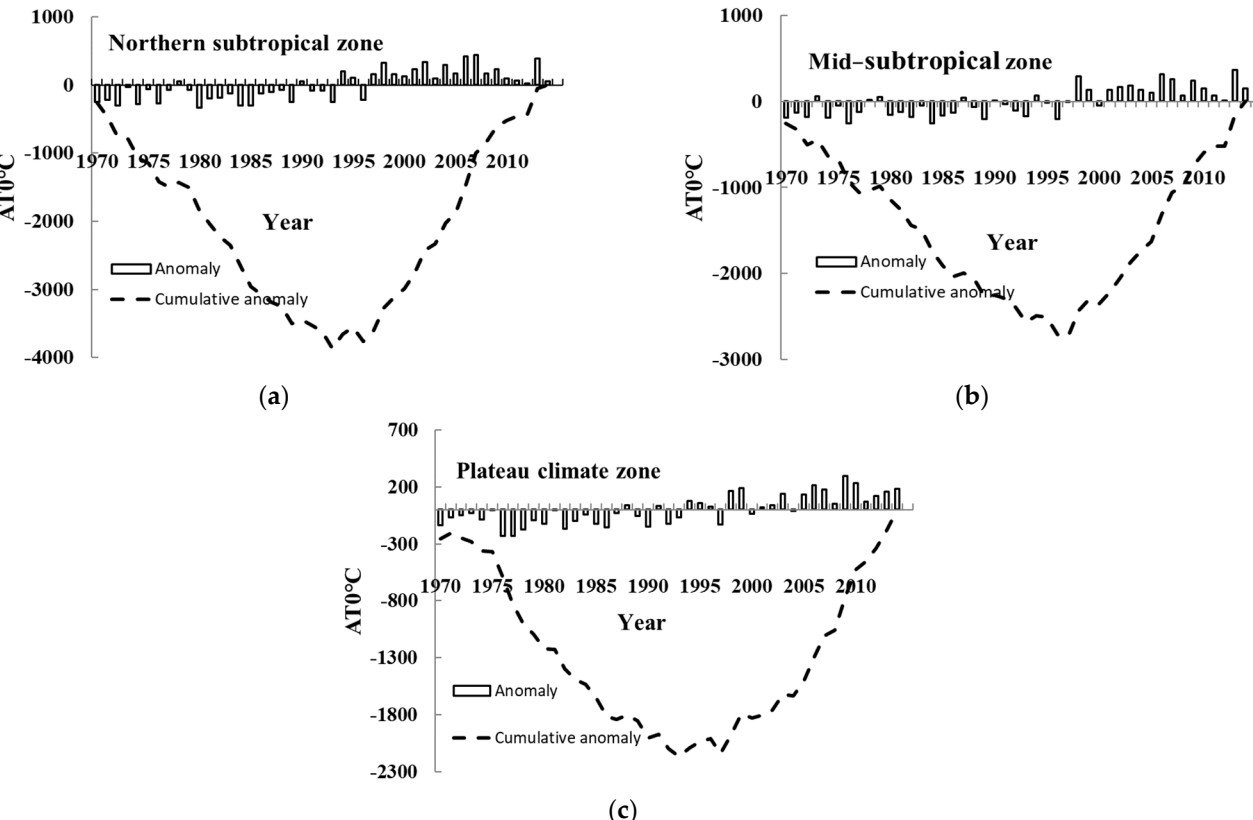

**Figure 3.** Anomalies and cumulative anomalies of an accumulated temperature of ≥0 °C in the northern subtropical zone (**a**), mid-subtropical zone (**b**), and plateau climate zone (**c**) of the Yangtze River Basin during 1970–2014.

From 1970 to 1997, the average accumulated temperature in the mid-subtropical zone was dominated by negative anomalies. After 1997, the average accumulated temperature

was mostly positive anomalies. Therefore, on a regional scale, the accumulated temperature of $\geq 0\,^{\circ}$C in the mid-subtropical zone had abrupt changes around 1997.

(2)  Interannual variation in the initial day, the final day, and the continuous days of accumulated temperature $\geq 0\,^{\circ}$C

From the interannual variation trend in the initial day, the final day and the continuous days of an accumulated temperature of $\geq 0\,^{\circ}$C in the northern subtropical zone, mid-subtropical zone and plateau climate zone from 1970 to 2014 (Figure 4), the results show that: in the northern subtropical zone and plateau climate zone since 1970, the initial day, the final day, and the continuous days of an the accumulated temperature of $\geq 0\,^{\circ}$C showed trends of advancement, postponement, and extension, respectively. The trends in the northern subtropical zone were $-2.52$ ($p = 0.06$), 0.17 ($p = 0.21$), and 3.38 d/10a ($p = 0.01$), and the tendency rates of the plateau climate zone were $-2.12$ ($p < 0.001$), 1.71 ($p < 0.001$), 3.92 d/10a ($p < 0.001$).

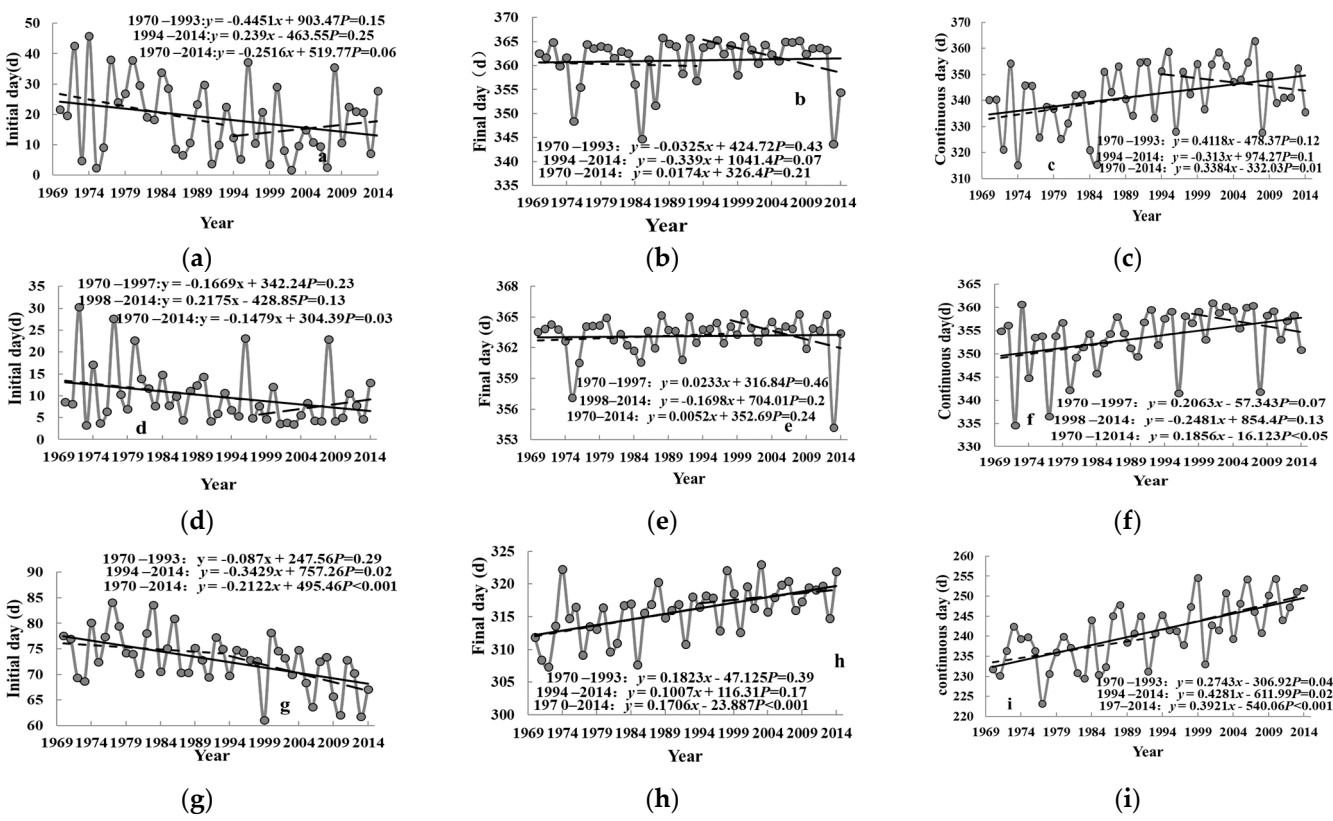

**Figure 4.** Interannual variation characteristics for the initial day, the final day, and the continuous days of an accumulated temperature of $\geq 0\,^{\circ}$C in the northern subtropical zone (**a–c**), mid-subtropical zone (**d–f**), and plateau climate zone (**g–i**) of the Yangtze River Basin during 1970–2014.

At the same time, two different periods can be observed: 1970–1993 and 1994–2014. During 1970–1993, the initial day of the northern subtropical zone advanced by $-4.45$ d/10a ($p = 0.15$), the final day advanced by $-0.33$ d/10a ($p = 0.29$), and the continuous days increased by 4.12 d/10a ($p = 0.12$); the plateau climate zone advanced by $-0.87$ d/10a ($p = 0.15$) for the initial day, 1.82 d/10a ($p = 0.39$) for the final day, and by 2.74 d/10a ($p = 0.04$) for the continuous days. However, during 1994–2014, affected by the rising trend of temperature, the initial day of the northern subtropical zone was delayed by 2.39 d/10a ($p = 0.25$), the final day advanced by $-3.39$ d/10a ($p = 0.07$), and the continuous days advanced by $-3.13$ d/10a ($p = 0.1$). The initial day of the plateau climate zone advanced by $-3.43$ d/10a ($p = 0.02$), the final day was delayed by 1 d/10a ($p = 0.17$), and the continuous days rose by 4.28 d/10a ($p = 0.02$). The interannual variation trends for the initial day, the

final day, and the continuous days in the northern subtropical zone and plateau climate zone all changed significantly around 1993.

During 1970–1997, the initial day of the mid-subtropical zone advanced by −1.67 d/10a ($p = 0.23$), the final day of the mid-subtropical zone was delayed by 0.23 d/10a ($p = 0.46$), and the continuous days of the subtropical zone was extended by 2.1 d/10a ($p = 0.07$); however, during 1998–2014, affected by the rising trend of temperature, the initial day of the mid-subtropical zone delayed by 2.18 d/10a ($p = 0.23$), the final day advanced by −1.7 d/10a ($p = 0.2$), and the continuous days were shortened by −2.48 d/10a ($p = 0.13$). The trends for interannual variation in the initial day, the final day, and the continuous days all changed significantly around 1997, which further proves that an accumulated temperature of ≥0 °C in the mid-subtropical zone had an abrupt change around 1997. In summary, the above analysis further explains that, in 1994, the accumulated temperature of ≥0 °C showed abrupt changes in the northern subtropical zone and plateau climate zone. An abrupt change also occurred in the mid-subtropical zone in 1997.

### 3.1.2. Spatial Characteristics

(1)　Spatial variation of accumulated temperature ≥0 °C

From 1970 to 2014, the interannual variation of an accumulated temperature of ≥0 °C in the Yangtze River Basin changed significantly on the spatial scale (Figure 5). Apart from a few stations, an accumulated temperature of ≥0 °C in the Yangtze River Basin presented a significant increasing trend. The 130 stations in the basin show increasing trends, of which 124 stations passed the significance level test ($p < 0.05$), accounting for 95.4% of the total number of stations in the basin, and only a few stations failed the significance level test. Although an accumulated temperature of ≥0 °C at most stations in the basin showed an increasing trend, a decreasing trend was present at one station, which was Guiyang station, with a downward trend of −57.2 °C/10a ($p < 0.05$). From a spatial perspective, the areas with a large increase in accumulated temperatures of ≥0 °C were mainly distributed in the Jialing River Basin, Dongting Lake Basin, Hanshui Basin, Poyang Lake Basin, Taihu Lake Basin, and the mainstream area of the lower reaches and the middle reaches.

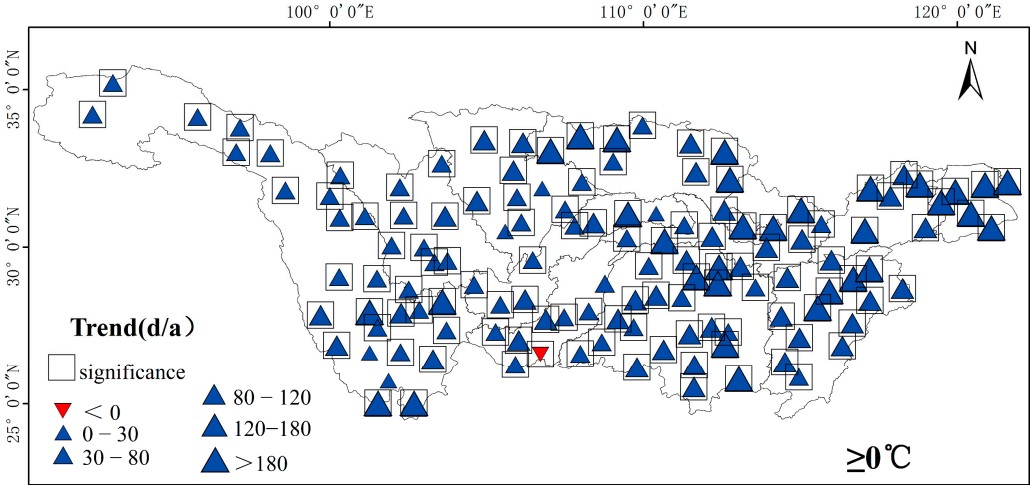

**Figure 5.** Spatial characteristics of interannual variation of an accumulated temperature of ≥0 °C for the Yangtze River Basin during 1970–2014. Squares indicate passing the 0.05 significance level, the same as below.

(2)　Spatial variation in the initial day, the final day and the continuous days of accumulated temperature ≥0 °C

From 1970 to 2014, the initial day of an accumulated temperature of ≥0 °C in the Yangtze River Basin showed an obvious advancing trend (Figure 6). Except for the initial day of Guiyang Station, which showed a postponed trend, all of the stations in the basin

presented advancing trends, which accounted for 99.2% of the stations. Among them, 35.4% of the stations passed the significance level test ($p < 0.05$) with an upward trend of $-0.2$–$-0.4$ d/10a, and the areas with large increases were concentrated in the Wujiang River Basin, the middle reaches of the mainstream, the Dongting Lake Basin, the Poyang Lake Basin and the Taihu Lake Basin. A postponement trend for the final day is obvious. In the study area, 103 stations show upward trends, accounting for 78.6% of the basin stations, of which 14.6% of the stations passed the significance level ($p < 0.05$), though the trend is relatively weak. By comparison, 28 stations have a downward trend, and most of the areas with a relatively concentrated decline are located in the Yunnan-Guizhou Plateau, the southern part of the Minjiang and Tuojiang rivers and the upper mainstream area.

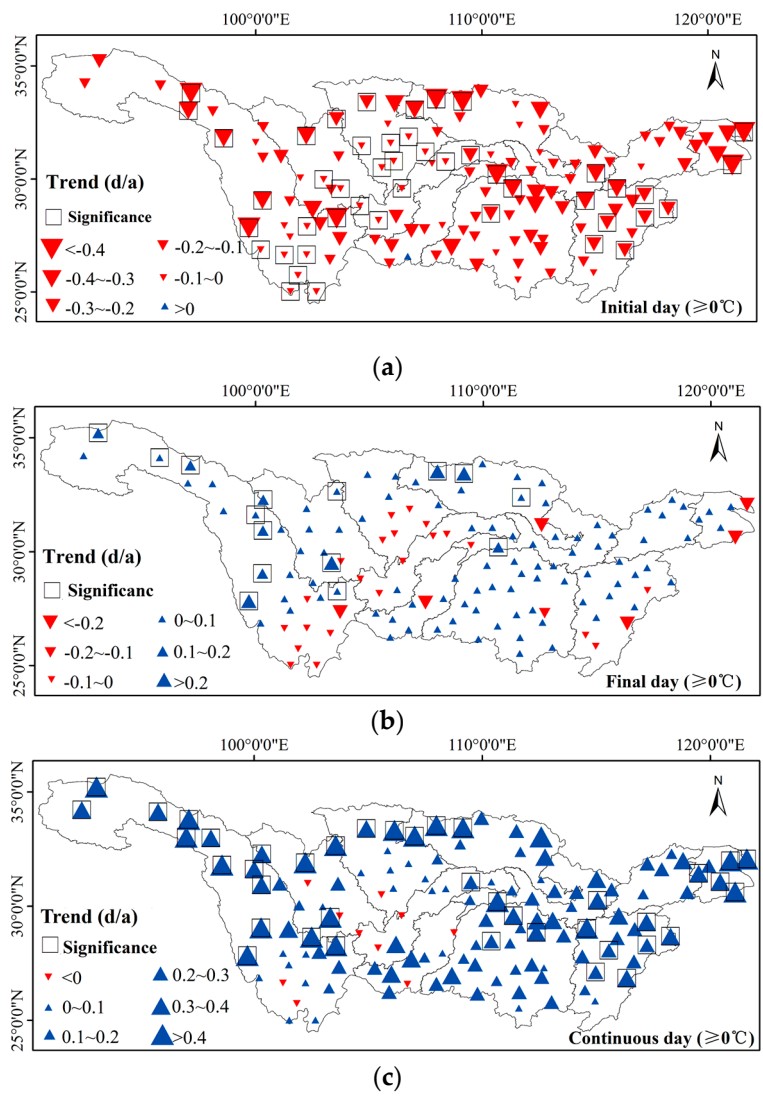

**Figure 6.** Spatial variation characteristics of the initial day (**a**), the final day (**b**), and the continuous days (**c**) of an accumulated temperature of $\geq 0$ °C for the Yangtze River Basin during 1970–2014.

The continuous days in the basin presented an obvious increasing trend. There are 121 stations in the basin showing an upward trend, accounting for 92.4% of the basin, of which 33.1% of the stations passed the significance level test ($p < 0.05$); the increase ranged between 0.2and 0.4; and the main areas are distributed in the upper and middle reaches of the Jinsha River Basin, the Hanshui Basin, the Poyang Lake Basin, the Dongting Lake Basin, the lower mainstream area, and the Taihu Lake Basin. By contrast, the upper reaches of the main stream area and the Jialing River basin did not increase significantly or showed a slight decline.

*3.2. Temporal and Spatial Characteristics of Accumulated Temperature ≥5 °C*

3.2.1. Temporal Characteristics

(1)  Interannual variation of accumulated temperature ≥5 °C

The Yangtze River Basin has a vast area, special geographical location, complex and diverse topography, and rich types of underlying surfaces. Analyzing the temporal variation for an accumulated temperature of ≥5 °C from different climatic regions is necessary. Figure 7 shows the interannual variation characteristics of an accumulated temperature ≥5 °C on a regional scale, namely in the northern subtropical zone, mid-subtropical zone, and plateau climate zone from 1970 to 2014. The average accumulated temperature in the three climatic regions was 5392.4, 5649.6 and 1997.6 °C. From the perspective of the trend for interannual change, since 1970, an accumulated temperature of ≥10 °C in the three climatic regions showed an increasing trend on the whole, with a tendency rate of 122.6 ($p < 0.001$), 90.5 ($p < 0.001$), and 81.4 °C/10a ($p < 0.001$). Moreover, the increase in the northern subtropical zone is significantly higher than that in the mid-subtropical zone and plateau climate zone. The accumulated temperatures in the northern subtropical zone and the plateau climate zone were different for two different periods, which are 1970–1993 and 1994–2014.

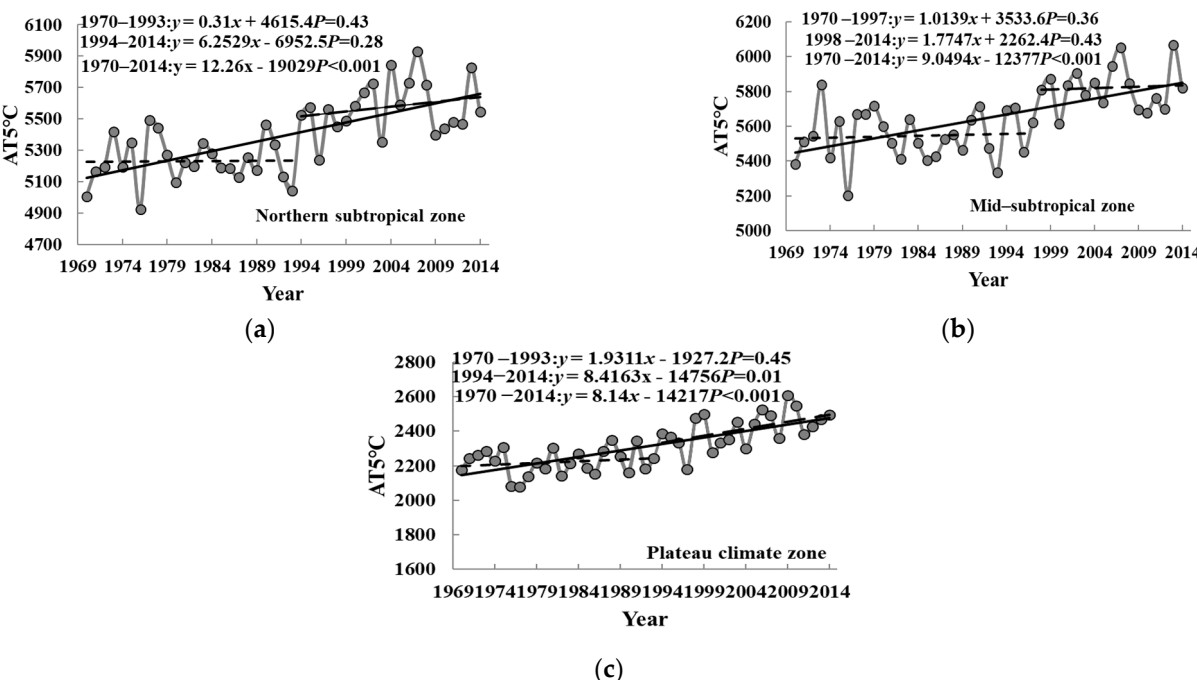

**Figure 7.** Interannual variation of an accumulated temperature of ≥5 °C in the northern subtropical zone (**a**), mid-subtropical zone (**b**), and plateau climate zone (**c**) of the Yangtze River Basin during 1970–2014.

Before the mid-1990s, the accumulated temperature of the three climatic regions increased with trends of 3.1 ($p = 0.43$), 10.1 ($p = 0.36$), and 19.3 °C/10a ($p = 0.45$); however, after the mid-1990s, affected by the rising trend of temperature, the accumulated temperature rose relatively sharply, with trends of 62.5 ($p = 0.28$), 177.5 ($p = 0.43$), and 84.2 °C/10a ($p = 0.01$) in the northern subtropical zone, mid-subtropical zone, and plateau climate zone, respectively. Figure 8 shows the anomalies in the accumulated temperature and accumulated anomalies for the northern subtropical zone, mid-subtropical zone, and plateau climate zone. The results indicate that the average accumulated temperature in the northern subtropical zone and plateau climate zone from 1970 to 1993 were dominated by negative anomalies. After 1993, the average accumulated temperature was dominated by positive anomalies. On the regional scale, the accumulated temperature of ≥0 °C in the

northern subtropical zone and the plateau climate zone showed abrupt changes around 1993.

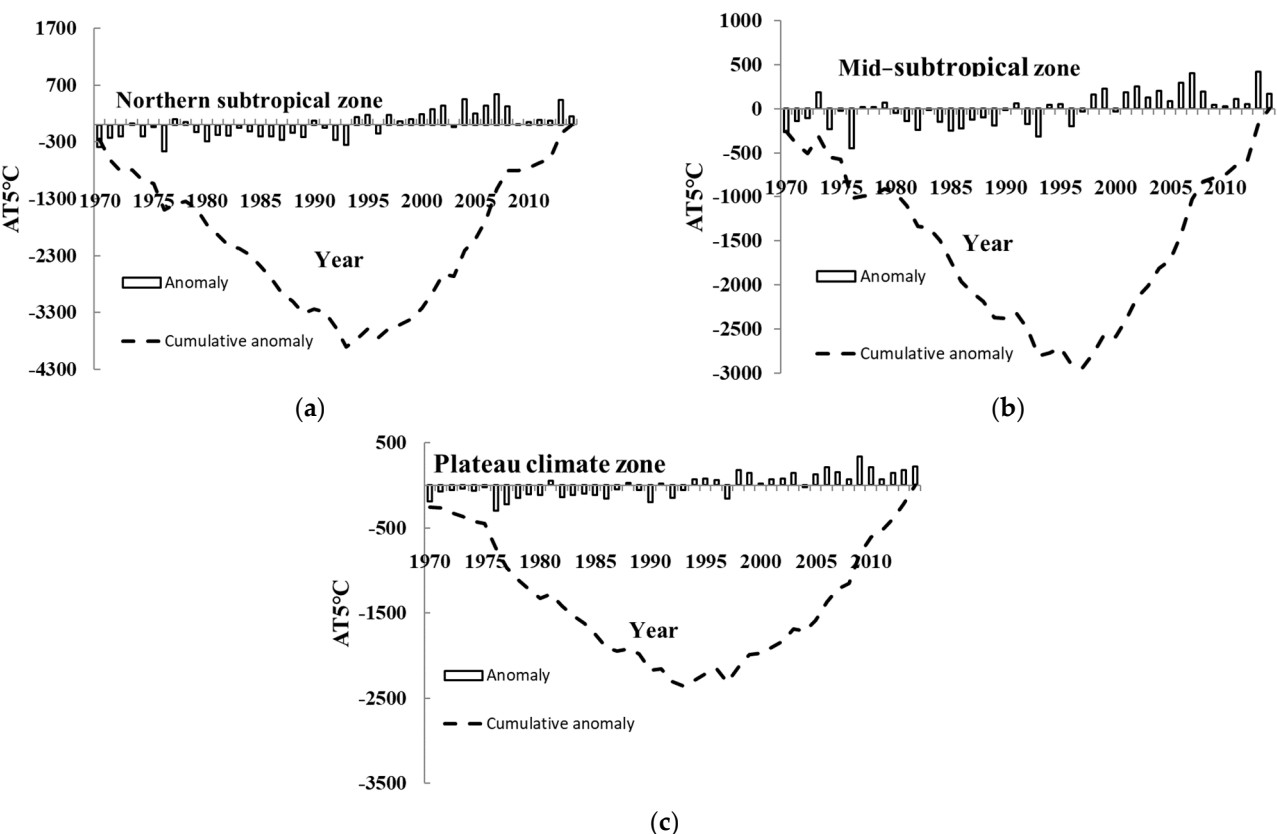

**Figure 8.** Anomaly and cumulative anomaly of an accumulated temperature of ≥5 °C in the northern subtropical zone (**a**), mid-subtropical zone (**b**), and plateau climate zone (**c**) of the Yangtze River Basin during 1970–2014.

From 1970 to 1997, the average accumulated temperature in the mid-subtropical zone was dominated by negative anomalies. After 1997, the average accumulated temperature was dominated by positive anomalies. Therefore, a conclusion can be drawn that, on the regional scale, the mid-subtropical zone showed an abrupt change for an accumulated temperature of ≥0 °C in 1997.

(2) Interannual variation in the initial day, the final day, and the continuous days of accumulated temperature ≥5 °C

The trends for interannual variation in the initial day, the final day, and the continuous days of an accumulated temperature of ≥5 °C in the northern subtropical zone, mid-subtropical zone, and plateau climate zone from 1970 to 2014 are shown in Figure 9.

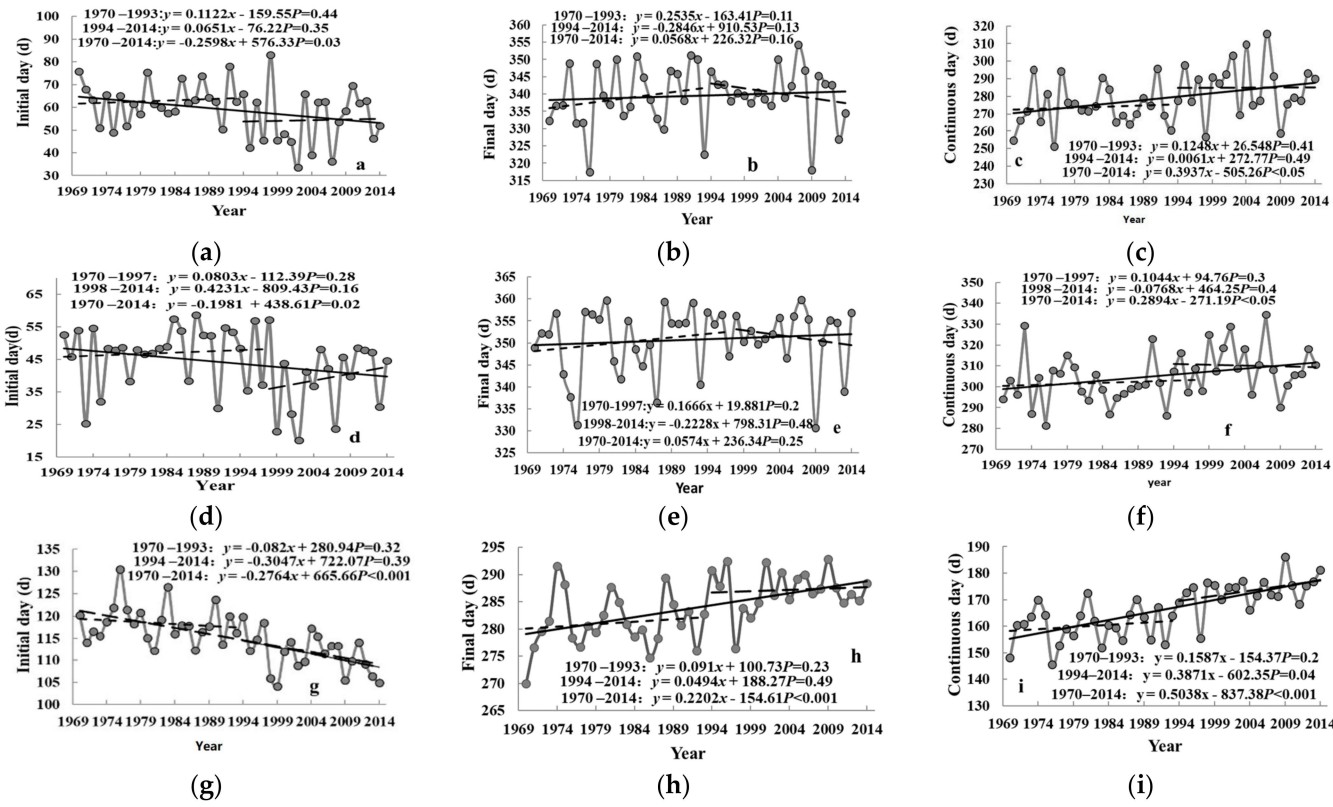

**Figure 9.** Interannual variation characteristics of the initial day, the final day, and the continuous days of an accumulated temperature of ≥5 °C in the northern subtropical zone (**a**–**c**), mid-subtropical zone (**d**–**f**), and plateau climate zone (**g**–**i**) of the Yangtze River Basin during 1970–2014.

The trends in the northern subtropical zone were −2.59 ($p$ = 0.03), 0.57 ($p$ = 0.16), and 3.94 d/10a ($p < 0.05$), and the trends of the plateau climate zone were −2.76 ($p < 0.001$), 2.2 ($p < 0.001$), and 5.03 d/10a ($p < 0.001$). Two different periods can be found: 1970–1993 and 1994–2014. During 1970–1993, the initial day in the northern subtropical zone was delayed by 1.12 d/10a ($p$ = 0.44), the final day was delayed by 2.54 d/10a ($p$ = 0.11), and the continuous days were extended by 1.25 d/10a ($p$ = 0.41). In the plateau climate zone, the initial day advanced by −0.82 d/10a ($p$ = 0.28), the final day was delayed by 0.91 d/10a ($p$ = 0.23), and the continuous days were extended by 1.59 d/10a ($p$ = 0.2). However, from 1994 to 2014, affected by the rising trend of temperature, the initial day of the northern subtropical zone was delayed by 0.65 d/10a ($p$ = 0.35), the final day advanced by 2.8 d/10a ($p$ = 0.137), and the continuous days increased by 0.06 d/10a ($p$ = 0.49).

In the mid-subtropical zone, since 1970, an accumulated temperature of ≥5 °C for the initial day, the final day, and the continuous days showed advanced, postponed, and increasing trends, respectively. The trends for changes in the mid-subtropical area were −2 ($p$ = 0.02), 0.57 ($p$ = 0.25), and 2.89 d/10a ($p < 0.05$). Two different periods were also found: 1970–1997 and 1998–2014. During 1970–1997, the initial day of the mid-subtropical zone advanced by 0.8 d/10a ($p$ = 0.28), the final day was postponed by 1.67 d/10a ($p$ = 0.2), and the continuous days were extended by 1 d/10a ($p$ = 0.3), while during 1998–2014, affected by the rising trend of temperature, the initial day in the mid-subtropical zone was delayed by 4.23 d/10a ($p$ = 0.16), the final day advanced by −2.2 d/10a ($p$ = 0.48), and the continuous days were shortened by −0.8 d/10a ($p$ = 0.4). The trends for interannual variation in the initial day, the final day, and the continuous days all changed significantly around 1997, which further proves that an accumulated temperature of ≥5 °C in the mid-subtropical zone presented a sudden change around 1997.

The above analysis further elaborates that, in 1994, an accumulated temperature of ≥5 °C showed abrupt changes in the northern subtropical zone and plateau climate zone from 1970 to 2014 and that the mid-subtropical zone showed a sudden change in 1997.

### 3.2.2. Spatial Characteristics

(1)    Spatial variation of accumulated temperature ≥5 °C

From 1970 to 2014, the interannual variation in an accumulated temperature of ≥5 °C in the Yangtze River Basin changed significantly in space (Figure 10). Except for a few stations, an accumulated temperature of ≥5 °C in the Yangtze River Basin showed a significant increasing trend. In the basin, 130 stations showed increasing trends, of which 118 stations passed the significance level test ($p < 0.05$), accounting for 90.1% of the total number of stations in the basin, and only a few stations failed the significance level test. Although an accumulated temperature of ≥5 °C in most stations showed an increasing trend, there was also a decreasing trend at one station: the Guiyang station with a decreasing trend of −75.1 °C/10a ($p < 0.05$). From a spatial point of view, areas with large increases in an accumulated temperature of ≥5 °C were mainly distributed in the upper main stream of the Jinsha River Basin, the Jialing River Basin, the Dongting Lake Basin, the Hanshui Basin, the Poyang Lake Basin, the Taihu Lake Basin, and the lower and middle reaches of the main stream.

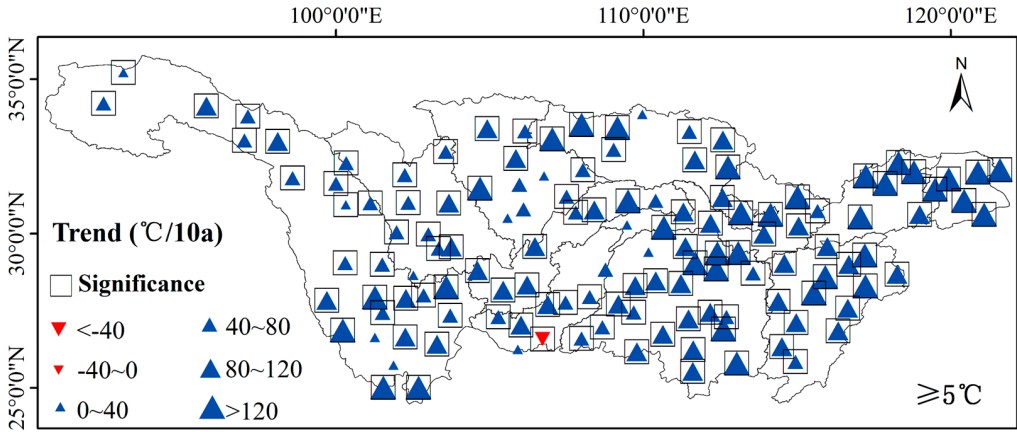

**Figure 10.** Spatial characteristics of interannual variation for an accumulated temperature of ≥5 °C of the Yangtze River Basin during 1970–2014.

(2)    Spatial variation in the initial day, the final day, and the continuous days of an accumulated temperature ≥5 °C

From 1970 to 2014, the initial day of an accumulated temperature of ≥5 °C in the Yangtze River Basin showed an obvious advancing trend (Figure 11). Except for the four stations in Yuexi, Macheng, Qianxi, and Anshun (all four stations passed the significance level test), which showed a postponed trend on the initial day, and Lijiang and Huize, which showed trends for no significant changes, all other stations showed an advancing trend, which accounted for 95.4% of the sites.

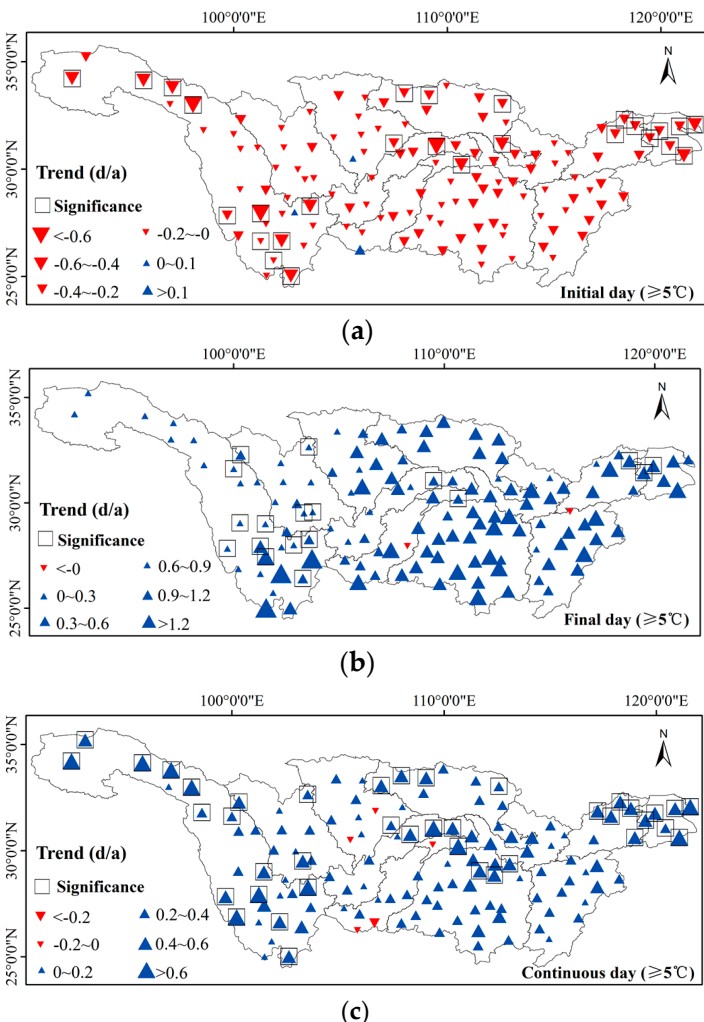

**Figure 11.** Spatial variation characteristics of the initial day (**a**), the final day (**b**), and the continuous days (**c**) of an accumulated temperature of ≥5 °C in the Yangtze River Basin during 1970–2014.

Among the sites, 20% passed the significance level test ($p < 0.05$), and the increase was between 0 and 1.2 d/a, and the areas with a large increase range were concentrated in Hanshui River Basin, the midstream mainstream area, Dongting Lake Basin, Poyang Lake Basin, Taihu Lake Basin, and the lower mainstream area. In the study area, a postponement trend of the final day was obvious, and 129 sites showed upward trends, accounting for 98.5% of the basin sites, of which 13.2% of the sites passed the significance level ($p < 0.05$), and the difference between the increasing trends was obvious. Areas with larger increases were mostly distributed in the basins south of the Yangtze River, such as the Dongting Lake Basin, Poyang Lake Basin, and the Yunnan-Guizhou Plateau, with trends of 0.93–0.9 d/10a.

Two stations showed a downward trend, which were located in the Wujiang River Basin and the lower reaches of mainstream area. The continuous days in the basin showed an obvious increasing trend, and 126 stations in the basin showed increasing trends, accounting for 96.2% of the stations, of which 33.1% of the stations passed the significance level test ($p < 0.05$), and the increase was mainly distributed between 0.2 and 0.6 d/a; the main areas with a large increase were concentrated in the upper and middle reaches of the Jinsha River Basin, Hanshui River Basin, Poyang Lake Basin, Dongting Lake drainage area, downstream mainstream area, and Taihu Lake basin. In the upper reaches of the main stream area and the Jialing River basin, the continuous days increased insignificantly or showed a slight downward trend.

*3.3. Temporal and Spatial Characteristics of an Accumulated Temperature ≥10 °C*

3.3.1. Temporal Characteristics

(1)  Interannual variation of an accumulated temperature ≥10 °C

Figure 12 shows the interannual variation characteristics of an accumulated temperature of ≥10 °C on the regional scale for the northern subtropical zone, mid-subtropical zone, and plateau climate zone from 1970 to 2014. The years of average accumulated temperature for the three climatic regions were 4919.4, 5031.9 and 1283 °C, respectively.

From the perspective of the trend for interannual change, since 1970, an accumulated temperature of ≥10 °C in the three climatic regions showed an overall increasing trend, and the trends for the change were 115.7 ($p < 0.001$), 92.5 ($p < 0.001$), and 78.9 °C/10a ($p < 0.001$). The increase in the northern subtropical zone was significantly higher than that in the mid-subtropical and plateau climate zone. The thermal accumulated temperature in the northern subtropical zone underwent variations for two different periods: 1970–1996 and 1997–2014, and that for the mid-subtropical zone and plateau climate zone were different for two different stages from 1970 to 1997 and from 1998 to 2014.

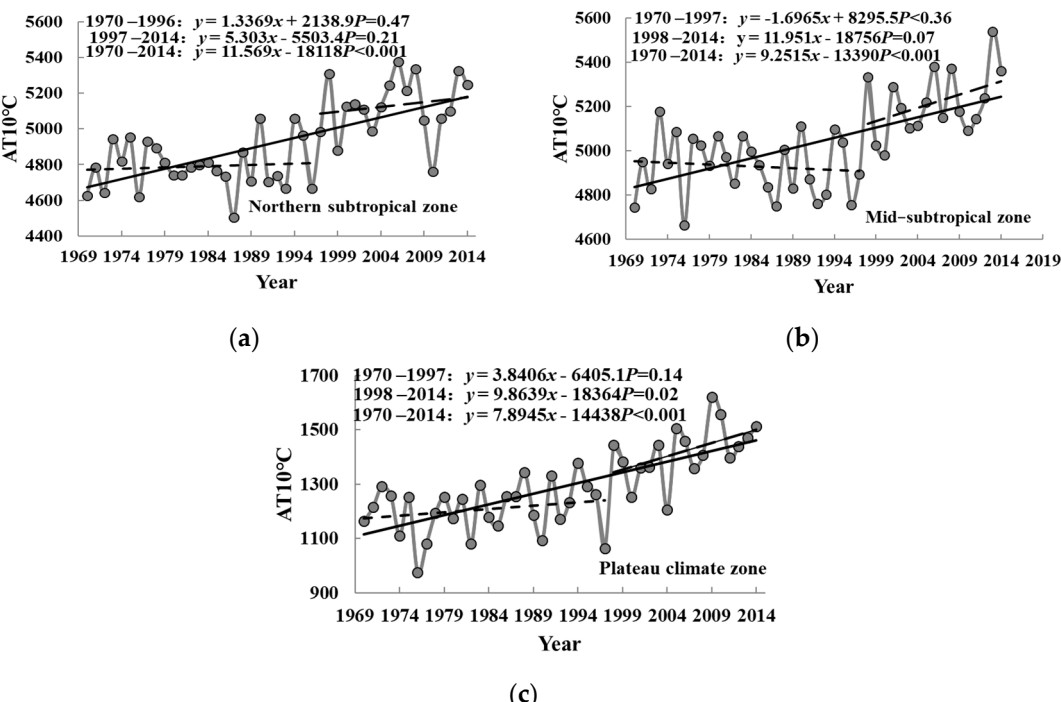

**Figure 12.** Interannual variation of an accumulated temperature of ≥10 °C in the northern subtropical zone (**a**), mid-subtropical zone (**b**), and plateau climate zone (**c**) of the Yangtze River Basin during 1970–2014.

Before the middle and late 1990s, the accumulated temperature in the three climatic regions increased or decreased with trends of 13.4 ($p = 0.21$), −17 ($p = 0.07$), and 38.4 °C/10a ($p = 0.14$), and after the mid to late 1990s, affected by the rising trend of temperature [17], the accumulated temperature increased sharply, with trends of 53 ($p = 0.21$), 120 ($p = 0.07$), and 98.6 °C/10a ($p = 0.02$). Figure 13 calculates the accumulated temperature anomalies and accumulated anomalies in the northern subtropical zone, mid-subtropical zone, and plateau climate zone. The results show that from 1970 to 1996, the average accumulated temperature in the northern subtropical zone is dominated by negative anomalies. After 1996, the average accumulated temperature was dominated by positive anomalies. On a regional scale, a sudden change occurred in the accumulated temperature in the northern subtropical zone around 1997. From 1970 to 1997, the average accumulated temperature in the mid-subtropical and plateau climate zone was dominated by negative anomalies.

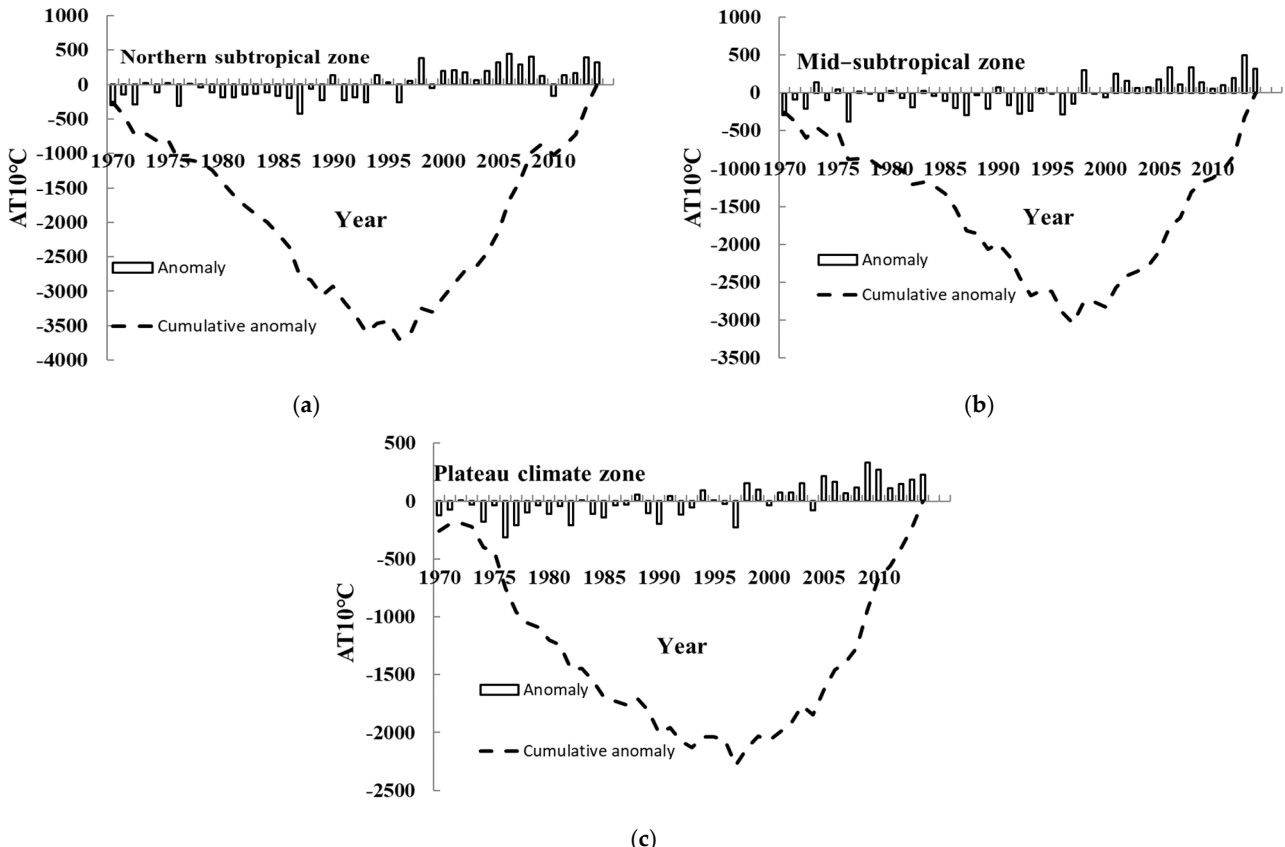

**Figure 13.** Anomaly and cumulative anomaly of an accumulated temperature of ≥10 °C in the northern subtropical zone (**a**), mid-subtropical zone (**b**), and plateau climate zone (**c**) of the Yangtze River Basin during 1970–2014.

After 1997, the average accumulated temperature was dominated by positive anomalies and the accumulated temperature in the mid-subtropical zone and plateau climate zone showed abrupt changes around 1998 on the regional scale.

(2)　Interannual variation in the initial day, the final day, and the continuous days of an accumulated temperature ≥10 °C

The trend for interannual variation in the initial day, the final day, and the continuous days of an accumulated temperature of ≥10 °C from 1970 to 2014 in the northern subtropical zone, mid-subtropical zone, and plateau climate zone can be seen in Figure 14.

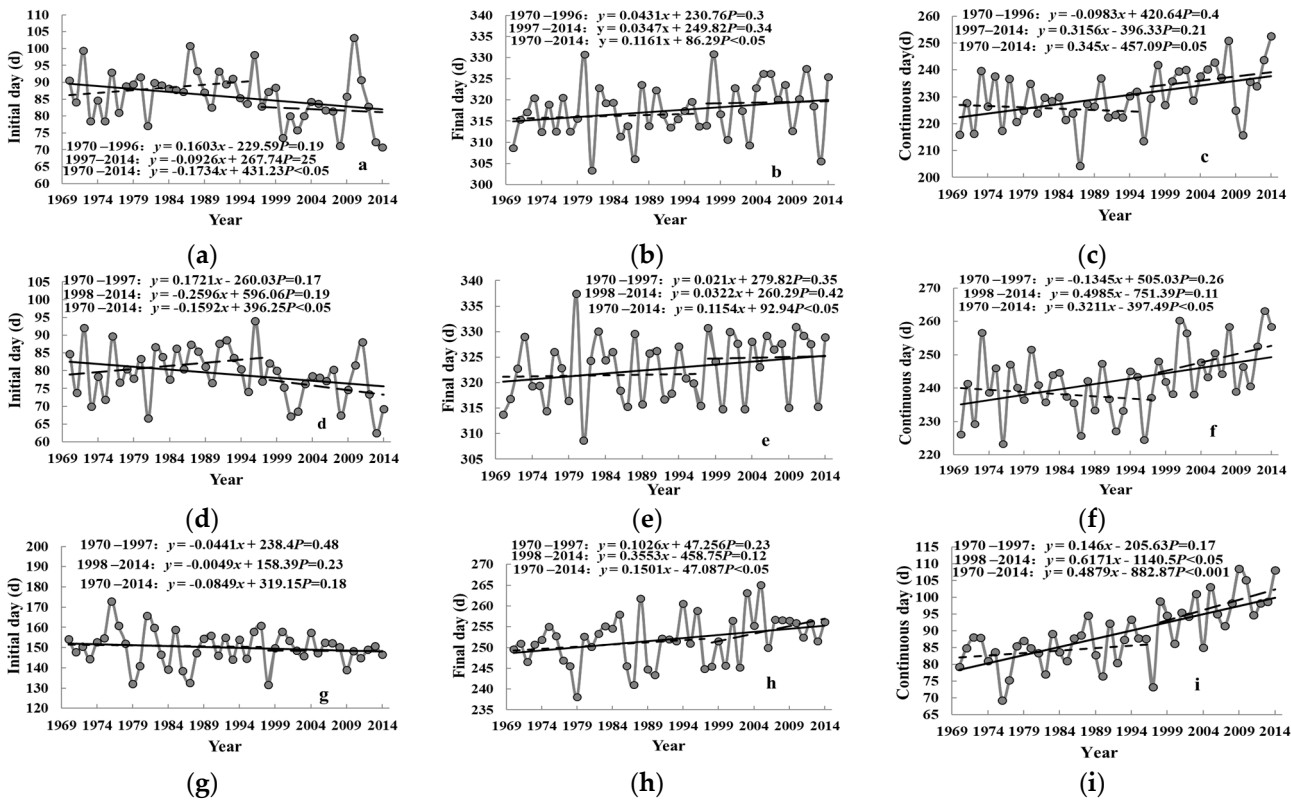

**Figure 14.** Interannual variation characteristics of the initial day, the final day, and the continuous days of an accumulated temperature of ≥10 °C in the northern subtropical zone (**a**–**c**), mid-subtropical zone (**d**–**f**), and plateau climate zone (**g**–**i**) of the Yangtze River Basin during 1970–2014.

The initial day, the final day, and the continuous days of an accumulated temperature of ≥10 °C showed trends of advancement, delay, and increase, respectively, and their change tendency rates were −1.73 ($p < 0.05$), 1.16 ($p < 0.05$), and 3.45 d/10a ($p < 0.01$). Two different periods were also found: 1970–1996 and 1997–2014. Between 1970 and 1996, the initial day and the final day were delayed by 1.6 ($p = 0.19$) and 0.43 d/10a ($p = 0.3$), respectively. The continuous days were shortened by 0.98 d/10a ($p = 0.4$); during 1997–2014, affected by the rising trend of temperature, the initial day advanced by 0.93 d/10a ($p = 0.25$), the final was day postponed by 0.35 d/10a ($p = 0.34$), and the continuous days were extended by 3.16 d/10a ($p = 0.21$). Their trends in the mid-subtropical were −1.6 ($p < 0.05$), 1.15 ($p < 0.05$), and 3.21 d/10a ($p < 0.05$), and in the plateau climate zone were −0.85 ($p = 0.18$), 1.5 ($p < 0.05$), and 4.88 d/10 a ($p < 0.001$). Two different periods were also found: 1970–1997 and 1998–2014.

During 1970–1997, the initial day of the mid-subtropical zone was delayed by 1.72 d/10a ($p = 0.17$), and the plateau climate zone was delayed by 0.4 d/10a ($p= 0.48$). The final days in the mid-subtropical and plateau climate zone were delayed by 0.21 ($p = 0.35$) and 1.03 d/10a ($p = 0.28$), respectively. In terms of the continuous days, the mid-subtropical zone was shortened by −1.35 d/10a ($p = 0.26$), while the plateau climate zone increased by 1.46 d/10a ($p = 0.19$). During 1998–2014, affected by the rising trend of temperature, the initial day in the mid-subtropical zone advanced by 2.6 d/10a ($p = 0.19$), the plateau climate zone advanced by 0.05 d/10a ($p = 0.23$), and the final days in these regions were postponed by 0.32 ($p = 0.42$) and 3.55 d/10a ($p = 0.12$), respectively. The interannual variation trends of the initial day, the final day, and the continuous days all changed significantly around 1998, which further proves that an accumulated temperature of ≥10 °C in the mid-subtropical and plateau climate zone showed abrupt changes around 1998.

In summary, the above analysis further shows that an accumulated temperature of ≥10 °C from 1970 to 2014 showed abrupt change in the northern subtropical zone in 1997,

and showed a sudden change in the mid-subtropical zone and plateau climate zone in 1998.

### 3.3.2. Spatial Characteristics

(1)    Spatial variation of an accumulated temperature ≥10 °C

Figure 15 shows that the interannual variation for an accumulated temperature of ≥10 °C in the Yangtze River Basin has obvious spatial differences. Except for a few stations, an accumulated temperature of ≥10 °C in the Yangtze River Basin showed an obvious increasing trend. Among them, 101 stations passed the significance level test ($p < 0.05$), accounting for 77% of the total number of stations in the basin. Only the Tuotuohe and Qingshuihe stations within the area did not pass the significance test; although the accumulated temperature of most sites increased, three stations (Huaping, Yuanmou, and Guiyang) had accumulated temperatures that showed a downward trend. Among them, the Guiyang station was the most significant, decreasing at a trend of 73.05 °C/10a ($p < 0.01$).

Areas with an accumulated temperature stable at 10 °C and increasing accumulated temperatures were distributed in the middle and lower reaches of the Yangtze River, which were east to the Hanzhong-Fengjie-Wufeng-Jishou-Wugang-Daoxian line, as well as at several sites in the Sichuan Basin, the Yunnan-Guizhou Plateau, and the Qinghai Plateau. Remarkably, the increased accumulated temperature was abnormally high with obvious spatial discontinuity for Muli (Sichuan Prov.), Leibo (Sichuan Prov.), Kunming (Yunnan Prov.), Fengjie (Chongqing City), and Wufeng (Hubei Prov.) stations.

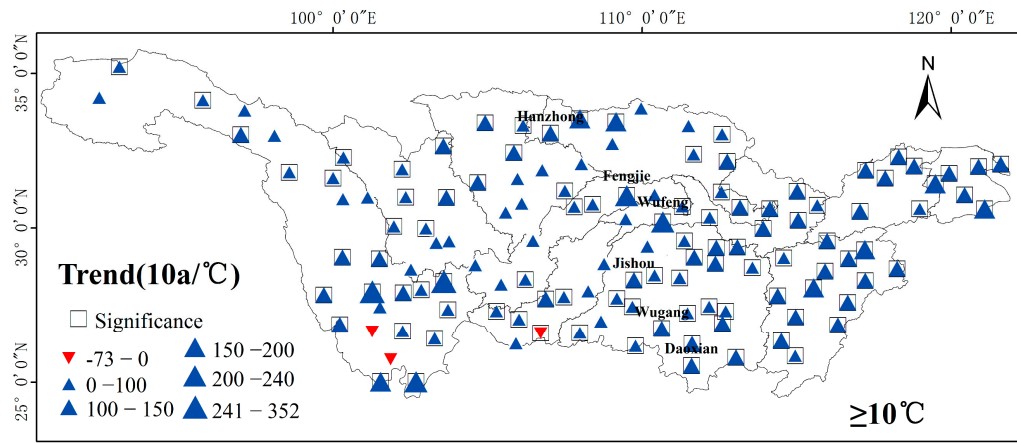

**Figure 15.** Spatial characteristics of the interannual variation of an accumulated temperature of ≥10 °C of the Yangtze River Basin during 1970–2014.

(2)    Spatial variation in the initial day, the final day, and the continuous days of an accumulated temperature ≥10 °C

From 1970 to 2014, the initial day of an accumulated temperature of ≥10 °C in the Yangtze River Basin showed an obvious advancing trend (Figure 16). Except for stations in Wudaoliang, Tuotuohe, Qianxi, and Anshun showing postponed trends in the initial day, all others showing an advancing trend, which accounted for 80.4%, and seven of them passed the significance level test ($p < 0.05$), with increasing ranges between −0.2 and −0.5 d/a. The areas with large increase ranges were distributed in the middle reaches of the mainstream area and Dongting Lake Basin, Poyang Lake Basin, Taihu Lake Basin, and the lower reaches of the mainstream area. A postponement trend of the final day is obvious.

In the study, 127 stations showed upward trends, accounting for 96.9% of the basin sites, of which 7% of the sites passed the significance level ($p < 0.05$), and the upward trend is obvious. The areas with significant upward trends were mostly concentrated in

the Jinsha River Basin and the Yunnan-Guizhou Plateau, with a trend of 0.8–1.6 d/a. The continuous days in the basin showed an obvious increasing trend. In the basin, 128 stations presented increasing trends, accounting for 97.7%, of which 25.7% of the stations passed the significance level test ($p < 0.05$), and the increase was mainly concentrated in 0.3–0.7 d/a. These were mainly concentrated in the upper and middle reaches of the Jinsha River Basin, the Poyang Lake Basin, the lower mainstream area, and the Taihu Lake Basin. The upper reaches of the main stream area increased insignificantly or showed a slight downward trend.

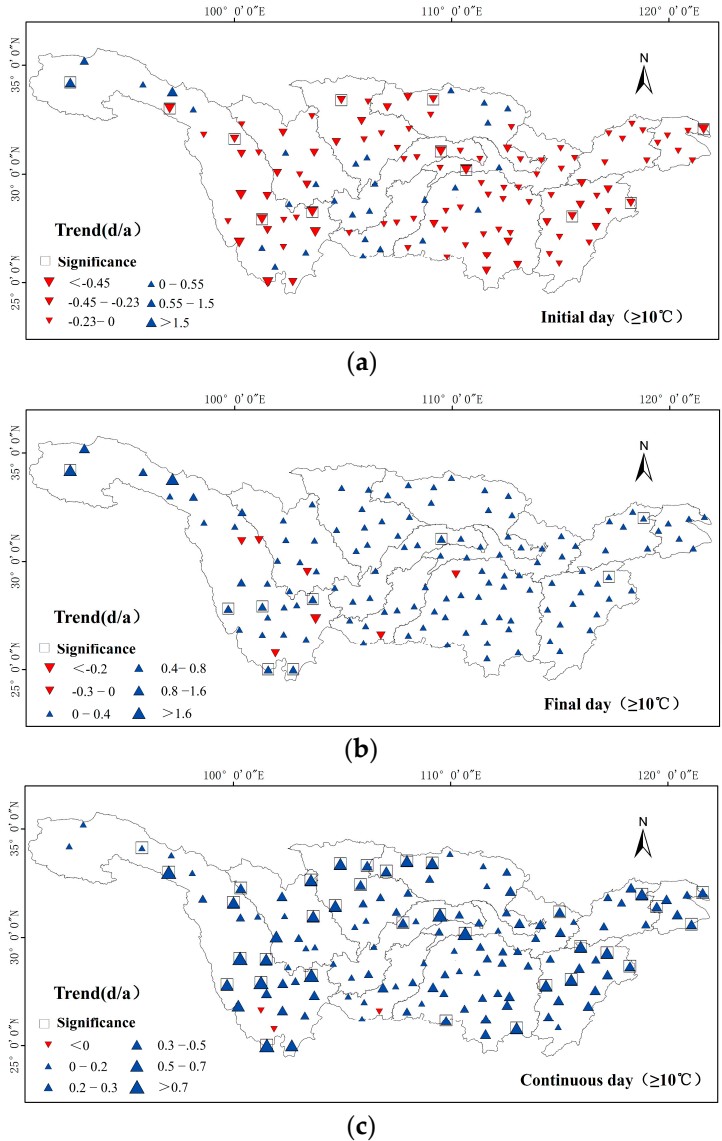

**Figure 16.** Spatial characteristics of the interannual variation of the initial day (**a**), the final day (**b**), and the continuous days (**c**) of an accumulated temperature of ≥10 °C in the Yangtze River Basin from 1970 to 2014.

## 4. Discussion

Accumulated temperatures of ≥0, ≥5, and ≥10 °C showed abrupt changes in the mid−1990s, in Yangtze River Basin. The temperature increased significantly in the period after the abrupt change in temperature. In the 1990s, the world entered a stage of warming hiatus. At present, the mechanism that created the warming hiatus is related to changes, such as anthropic aerosols [18], volcanoes [19], solar variations [20], and sea surface temperature [21,22]. The annual mean temperature in the Yangtze River Basin

increased with a trend of 0.275 °C/10a. Average accumulated temperatures of >0, >5, and >10 °C increased by more than 96 °C/10a, and the average continuous days increased by 3.6 d/10a. The accumulated temperature and continuous days increased by 349 °C/a and 13 d/a for each 1° increase in mean temperature in the Yangtze River Basin. Due to global warming, the number of heat sources in the Yangtze River Basin will continue to increase. Due to differences in the natural environmental background, human activities, and other factors in different regions, the response time and amplitude of thermal resources to global warming show certain regional differences. A large number of studies have shown that the influence of the urban heat island effect on temperature in China does exist and that these regional characteristics.

In this study, the average and accumulated temperature increases in the middle and lower reaches of the Yangtze River have increased higher than those in other areas within the basin, which are particularly obvious in the lower reaches of the mainstream area and the Taihu Lake Basin. This is closely related to the strong human activities in the middle reaches, the lower reaches of the mainstream area, and the Taihu Lake Basin, and to the large scale of urban land use, and the high level of urbanization, indicating that the heat island effect caused by urbanization has a significant impact on the increase in air temperature and accumulated temperature, and that the growing season in the basin has been significantly prolonged, providing better thermal conditions for the area.

Studies have shown that as the accumulated temperature increases, the planting system gradually extends from low latitudes to high latitudes and from low-altitude areas to high-altitude areas [23]. This is very beneficial for adjusting the structure of crop varieties, extending the growth period of crops, and increasing yields. On the one hand, the Yangtze River Basin is an important area of grain production in China. An appropriate increase in planting areas using multi-cropping systems, such as wheat-cotton double-cropping, wheat-rice double-cropping, rape-rice double-cropping, wheat-corn-rice triple-cropping, and wheat (canola)-rice-rice triple-cropping, have brought new opportunities for the adjustment of the agricultural industrial structure and the increase in grain production in this area [24]. On the other hand, the accumulated temperature increases, the growth period advances, and the germination and greening of plants in spring advances. Most of the Yangtze River Basin is located in the monsoon climate region and late spring cold periods, when the crops are susceptible to low-temperature damage, occur often. This poses new challenges to agricultural production in the Yangtze River Basin.

Compared with previous changes in the accumulated temperature of simple administrative units in this area [25], this paper analyzed the temporal and spatial change characteristics of accumulated temperatures of $\geq$0, $\geq$5, and $\geq$10 °C, within the entire region of the Yangtze River Basin from 1970 to 2014, which provides a more detailed analysis of the distribution characteristics of the initial day, the final day, and the accumulated temperature in the region, serving certain practical significance for understanding and guiding agricultural production in the region. However, a site-based analysis cannot fully reflect the distribution of heat sources in the entire basin. In the future, the application of remote sensing satellite data should be further harnessed.

## 5. Conclusions

(1) Since 1970, an accumulated temperature of $\geq$0 °C in the northern subtropical zone, mid-subtropical zone, and plateau climate zone has shown overall increasing trends, and the trends were122 ($p < 0.001$), 87.7 ($p < 0.001$), and 75.3 °C/10a ($p < 0.001$), respectively, with an increase in the northern subtropical zone being significantly higher than that in the mid-subtropical zone and plateau climate zone. The northern subtropical zone showed a sudden change in 1997, and the mid-subtropical climate and the plateau climate zone showed sudden changes in 1994. In the basin, the initial day generally advanced, the final day was delayed, and the continuous days increased.

(2) Since 1970, an accumulated temperature of $\geq 5$ °C in the northern subtropical zone, mid-subtropical zone, and plateau climate zone generally increased, with tendency rates of 122.6 ($p < 0.001$), 90.5 ($p < 0.001$), and 81.4 °C/10a ($p < 0.001$), with the increase in the northern subtropical zone being significantly higher than that in the mid-subtropical zone and plateau climate zone. The increase in the northern subtropical zone was significantly higher than that in the mid-subtropical and plateau climate zone. The abrupt change in the north subtropical zone was in 1997, and that in the mid-subtropical and plateau climate zone was in 1994. In the basin, the initial day generally advanced, the final day was delayed, and the continuous days increased.

(3) Since 1970, an accumulated temperature of $\geq 10$ °C in the northern subtropical zone, mid-subtropical zone, and plateau climate zone generally increased, with a trend of 115.7 ($p < 0.001$), 92.5 ($p < 0.001$), and 78.9 °C/10a ($p < 0.001$), where the increase in the northern subtropical zone was significantly higher than that in the mid-subtropical zone and plateau climate zone. The accumulated temperature $\geq 10$ °C in the north subtropical zone showed an abrupt change in 1997, and the mid-subtropical and plateau climate zone showed an abrupt change in 1998. Except for a few stations, all stations in the study area showed a significant increase in accumulated temperatures of $\geq 10$ °C, but more obvious differences were present for different areas. The stations with large increases were mainly located in the middle and lower reaches of the Hanshui River Basin and the main stream area of the middle reaches, lower mainstream area, Poyang Lake Basin, Taihu Lake Basin, and other areas.

(4) The Yangtze River Basin should draw on the advantages and avoid disadvantages in the strategy to cope with climate change. With the extension of the growing season and the increase in heat resources, on the one hand, the advance in the germination time or rejuvenation time of forages and cold-loving crops and the delay in the end of the growth period can appropriately increase the development of animal husbandry. On the other hand, appropriate late-maturing varieties should be selected for crop variety breeding, first, to make full use of heat resources and to improve the quality of agricultural products and, second, to adjust the planting system and to improve the multiple cropping index to steadily increase agricultural output. The increase in heat sources in the Yangtze River Basin brings new opportunities for adjusting the agricultural industrial structure and increasing farmers' income in the Yangtze River Basin.

**Author Contributions:** Conceptualization, Guangxun Shi and Peng Ye; Data curation, Xianwu Yang; Formal analysis, Guangxun Shi; Investigation, Guangxun Shi and Peng Ye; Methodology, Guangxun Shi; Project administration, Xianwu Yang; Validation, Xianwu Yang; Visualization, Guangxun Shi; Writing—original draft, Guangxun Shi; Writing—review and editing, Peng Ye and Guangxun Shi. All authors have read and agreed to the published version of the manuscript.

**Funding:** This research was funded by the National Natural Science Foundation of China (grant no.41701449).

**Institutional Review Board Statement:** Not applicable.

**Informed Consent Statement:** Not applicable.

**Data Availability Statement:** The data presented in this study are available on request from the corresponding author.

**Acknowledgments:** The authors thank Mingjun Ding of Jiangxi Normal University for his critical reviews and constructive comments.

**Conflicts of Interest:** The authors declare no conflict of interest.

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
