# Peer review of "Spatio-Temporal Variation Analysis of the Biological Boundary Temperature Index Based on Accumulated Temperature: A Case Study of the Yangtze River Basin"

_ijgi, doi:10.3390/ijgi10100675_

Round 1
Reviewer 1 Report
Please check the document attached.

Author Response
Response to Reviewer 1 Comments
This manuscript describes statistical research for the accumulated temperature in Yangtze RiverBasin from 1970 to 2014. This index is very important for the agricultural production, which is a hot topic in agronomy, geography and the related fields. For my opinion, however, the method used and main conclusion in this manuscript is quite similar to the previous research (a good example is “The spatio-temporal change of active accumulated temperature ≥ 10℃ in Southern China from 1960 to2011”), so the innovate points of the research is not outstanding. Additionally in the current version the English writing should be significantly improved. As a general comment, consequently, I think this manuscript is not ready to be published in the journal.
Response: (1) We thank the reviewer for the valuable comment. Firstly, the article "the spatial temporary change of active accumulated temperature ≥ 10℃ in southern China from 1960 to 2011" is distributed in Guangdong, Guangxi, Yunnan and Fujian. The Yangtze River Basin is located in the north of the area (Guangdong, Guangxi, Yunnan and Fujian province). Secondly, this paper analyzes not only the accumulated temperature ≥10℃, but also the accumulated temperatures ≥ 0℃ and ≥5℃.
(2) We thank the reviewer for the valuable comment. We have used MDPI English Editing Services.
Detailed comments (the numbers below refer to the line numbers in the manuscript):
Point 1. L48: modify "exploration by scholars at home and abroad" as "which is explored by scholars worldwide", to make it more understandable.
Response 1: Thank you for pointing this out. We added this point in revised manuscript and the detailed revision can be found in Line 55-56, Page2.
Point 2. L50: replace "Researches" by "Research"
Response 2: Thank you for pointing this out. We added this point in revised manuscript and the detailed revision can be found in Line 58, Page2.
Point 3. L78: In this part the knowledge of the study area is missing (the total area of the place, the seasonal character, etc.), especially how you tell the difference between the three zones?
Response 3: We thank the reviewer for the valuable comment. We added this point in revised manuscript and the detailed revision can be found in Line 89-96, Page2 and Line 97-105, Page3.
The Yangtze River Basin (90°33'~122°25'E, 24°30'~35°45'N) (Figure 1) Including the vast area of which the main stream and tributaries of the Yangtze River. It is the third largest basin in the world, with a total area of 1.8 million square kilometers, accounting for 18.8% of China. The Yangtze River Basin spans 19 provincial administrative units of China's three major economic zones (Eastern Economic Zone, central economic zone and Western Economic Zone). In addition, the Yangtze River Basin is also an important commodity grain base in China. On the whole, the annual average temperature in the Yangtze River Basin is high in the East, low in the west, high in the South and low in the north. Except for some high-altitude areas such as the Western Sichuan Plateau and the source of the Yangtze River, most areas of the basin belong to subtropical monsoon climate. The northern subtropical zone is located at 28 ° ~ 33 ° N, including the Hanshui River Basin, the middle and lower reaches of the Yangtze River and the Taihu Lake Basin, accounting for about 5.4% of the national land area. The middle subtropical zone is located in the Han River Basin, the middle and lower reaches of the Yangtze River, the south of the Taihu Lake Basin and the east of the Qinghai Tibet Plateau, accounting for about 16.5% of China's land area. Qinghai Tibet Plateau is mainly located in the plateau area west of 103 ° E. The division is mainly based on Zheng's research results.
Point 4. L88: modify this part as ", and the daily average temperature was used for the analysis. For the stations without this index, ...."
Response 4: We thank the reviewer for the valuable comment. We added this point in revised manuscript and the detailed revision can be found in Line 118-120, Page3.
Point 5 In Figure 2, how you find the breakpoint of the year (i.e. please explain the procedure for finding the breakpoint). Also, please illustrate the meaning of the dashed lines in each sub-plot.
Response5:
(1) We thank the reviewer for the valuable comment.
We use Cumulative anomaly to find the breakpoint.
Cumulative anomaly is a common method to judge the change trend directly from the curve. The cumulative anomaly curve shows an upward trend, indicating that the anomaly value increases, and a downward trend indicates that the anomaly value decreases. From the obvious ups and downs of the curve, we can judge the long-term significant evolution trend and continuous change, and even diagnose the approximate time of abrupt change. We added this point in revised manuscript and the detailed revision can be found in Line Line 148-160, Page4.
(2) We thank the reviewer for the valuable comment. The dotted line in each sub-plot indicates the change trend of accumulated temperature before and after of the abrupt change point.
Point 6. L125: That's very tricky...All the three p-values are higher than 0.05, which means that the changing trend is not significant. The conclusion, consequently, is not promising.
Response 6: We thank the reviewer for the valuable comment.
Firstly, the selection of the above time periods is determined according to the abrupt change points obtained from accumulated temperature accumulation anomaly. The cumulative abnormal curve can be drawn for trend and abrupt change analysis. The scientific consensus is that global temperature warming hiatus in the 1990s. Therefore, the accumulated temperature of the Yangtze River basin appeared a turning point and abrupt change during the 1990s. Secondly, there are 131 meteorological stations in the study area, and a small number of stations have not passed the significance level test of P≤0.05, so the significance level after regional average will be higher than P≤0.05.
Point 7. L154: Similar as above, the P values are always higher than 0.05, which means that the conclusion is not sigificant....
Response 7: We thank the reviewer for the valuable comment.
Firstly, the selection of the above time periods is determined according to the abrupt change points obtained from accumulated temperature accumulation anomaly. The cumulative abnormal curve can be drawn for trend and abrupt change analysis. The scientific consensus is that global temperature warming hiatus in the 1990s. Therefore, the accumulated temperature of the Yangtze River basin appeared a turning point and abrupt change during the 1990s. Secondly, there are 131 meteorological stations in the study area, and a small number of stations have not passed the significance level test of P ≤0.05, so the significance level after regional average will be higher than P≤0.05.
Point 8. L182: I don't know the causality relationship here: why the rising trend of temperature can advance the initial and the final day??
Response8: We thank the reviewer for the valuable comment. should replace "advanced" by "delayed". We added this point in revised manuscript and the detailed revision can be found in Line 247, Page7.
Point 9. In Figure 5 the size of different triangles is close to each other, especially the one showing 80~120and >120, so it is better to change the symbol.
Response 9: We agree with the reviewer's comment. We added this point in revised manuscript and the detailed revision can be found in Line 270-271, Page8.

Reviewer 2 Report
The paper is good, however I suggest some points to improve that.
- Too long, and too much information condensed into one article. I suggest to shorten that or make two articles from the material.
- There are missing fields like information on vegetation and cropping structures that may be influenced by climate change ("Besides, the heat resources have shown a significant increasing trend, which has guiding significance for the futureproduction and development of agriculture in this region").
- Within the conclusion part some suggestions would be welcome regarding the handling of this phenomena.
- A thorough grammatical revision would make a benefit.
Author Response
Response to Reviewer 2 Comments
Point 1. Too long, and too much information condensed into one article. I suggest to shorten that or make two articles from the material.
Response 1: We thank the reviewer for the valuable comment.
Shorten parts of the paper can be found in Line 169-171, Page4, Line 190-194, Page 5, Line 198-201, Page 5, Line 234-241, Page 7, Line 310-312, Page 9, Line 331-341, Page 10, Line 351-353, Page 11, Line 357-359, Page 12, Line 371-376, Page 12, Line 464-466, Page 15, Line 500-501, Page 17, Line 511-513, Page 17, Line 528-533, Page 17.
Point 2. There are missing fields like information on vegetation and cropping structures that may be influenced by climate change ("Besides, the heat resources have shown a significant increasing trend, which has guiding significance for the future production and development of agriculture in this region").
Response 2: We thank the reviewer for the valuable comment. We added this point in revised manuscript and the detailed revision can be found in Line 42-46, Page 1.
With the increase of heat resources in the Yangtze River Basin, appropriate late maturing varieties should be selected in variety breeding, so as to make full use of heat resources and improve the quality of agricultural products. Secondly, adjust the planting system and improve the multiple cropping index to steadily increase agricultural output. This brings new opportunities to adjust the structure of agricultural industry and increase farmers' income, in the Yangtze River basin.
Point 3. Within the conclusion part some suggestions would be welcome regarding the handling of this phenomena.
Response 3: We thank the reviewer for the valuable comment. We added this point in revised manuscript and the detailed revision can be found in Line 700-711, Page 21.
The Yangtze River Basin should draw on the advantages and avoid disadvantages in the strategy of coping with climate change. With the extension of the growing season and the increase of heat resources, on the one hand, the advance of the germination time or rejuvenation time of forages and cold loving crops and the delay of the stop growth period can appropriately increase the development of animal husbandry. On the other hand, appropriate late maturing varieties should be selected for crop variety breeding, so as to make full use of heat resources and improve the quality of agricultural products. Secondly, adjust the planting system and improve the multiple cropping index to steadily increase agricultural output. The increase of heat resources in the Yangtze River Basin brings new opportunities for adjusting the agricultural industrial structure and increasing farmers' income in the Yangtze River Basin.
Point 4. A thorough grammatical revision would make a benefit.
Response 4: We thank the reviewer for the valuable comment. We have used MDPI English Editing Services.

Reviewer 3 Report
Please see my attached comments.

Author Response
Response to Reviewer 3 Comments
Summary: This manuscript describes decadal signal in accumulated temperature based on daily average temperature data from 131 stations located in the various sub-regions of the Yangtze River Basin. As the manuscript aptly states, this region is important to Chinese grain production and Chinese food security; therefore, understanding the spatial variability in temperature and its evolution in time is critical for adapting agricultural practices to regional climate change. The article is well written and very clear. I recommend minor revisions and a few small suggestions. Please see my itemized comments below:
Minor Suggestions:
Point 1: I suggest describing the methodology in more detail. How has the temperature been processed? What qualifies a day as >0℃, >5℃, or >10℃? How is accumulated temperature calculated? Consider including an equation that describes accumulated temperature. I think this will improve the manuscript, and make it more readable.
Response 1: We agree with the reviewer's comment. We added this point in revised manuscript and the detailed revision can be found in Line 128-132, Page 3 and 133-147, Page4.
AT0, AT5 and AT10 are defined as the sum of daily mean temperatures above 0℃, 5℃and 10℃ in a continuous period of 1 year. If temperature values exceed 0℃, 5℃and 10℃ for five consecutive days, then the AT 0, AT5 and AT10 are calculated from the first day of those five days and extends until the final day above 0℃, 5℃ and 10℃.
Point 2: This manuscript does well reporting what was found, but could improve its interpretation of the findings. For example, in Line 125, 31.3℃/10a is the accumulated temperature in the northern subtropical zone. But what does that equate to? Does that mean that 3 days each year are much warmer than the average, or perhaps, sixty days per year are 1℃ warmer on average? How can you translate this number into something actionable? The same can be said for the analysis of initial-final-continuous days. A few instances of these interpretations would add emphasis to the findings.
Response 2: We thank the reviewer for the valuable comment.
There are many indicators to be considered, so it is analyzed from the whole Yangtze River Basin. We added this section to the discussion section. We added this point in revised manuscript and the detailed revision can be found in Line 620-625, Page20.
The annual mean temperature in the Yangtze River Basin increases with a trend of 0.275℃/10a. The average accumulated temperature of >0℃, > 5℃and >10℃increased by more than96℃/10a, and the average continuous days increased by 3.6d/10a.The accumulated temperature and continuous increased by 349℃/a and 13d/a for each 1°increase of mean temperature in the Yangtze River Basin.
Point 3: I would expect that the values for >0℃ to be the largest, followed by >5℃, and followed lastly by >10℃, but in the conclusions that is not the case. I am under the impression that these data are inclusive, suggesting that >0 o would include all data in both the other two categories, as well as data points from>0℃to <5℃. Can you comment as to why?
Response 3: We thank the reviewer for the valuable comment.
The accumulated temperature of >0℃, >5℃ and >10℃are defined as the sum of daily mean temperatures above 0℃, 5℃and 10℃in a continuous period of 1 year. Before the average temperature did not reach 0°. Due to the increase of temperature, the average temperature just exceeded 0°C, meeting the requirement that the average temperature exceeded 0°C for 5 consecutive days and the accumulated temperature>0°C increased. However, the increased accumulated temperature of these days has no effect on the increase of accumulated temperature of > 5℃ and > 10℃, because the accumulated temperature of >5℃ and>10℃ is the sum of the average temperature≥5℃ and≥10℃for five consecutive days. Similarly, with the increase of temperature and the increase of days >5℃ and >10℃, the accumulated temperature will also increase, but it has little effect on the change of accumulated temperature≥0℃.
Point 4: In many sections, the authors report on abrupt changes in the observed trends occurring in the mid-1990s. Can the authors speculate as to why? The global circulation, policy, a volcanic eruption, or changes in instruments are all possible.
Response 4: We agree with the reviewer's comment. We added this point in revised manuscript and the detailed revision can be found in Line 615-620, Page20.
The accumulated temperature of ≥0℃, ≥5℃ and ≥10℃had abrupt changes in the mid-1990s, in the Yangtze River Basin The temperature increased significantly in a period of time after the abrupt change of temperature. In the 1990s, the global has entered the stage of warming hiatus. At present, the explanation of the formation mechanism of warming hiatus is that it is related to changes such anthropic aerosols [18], volcanoes [19], solar variations [20], and sea surface temperature [21,22].
Specific Comments:
Point 1: Lines 125–127: Are the numbers from before the mid-1990s and during the mid-1990s supposed to be the same? I would not guess so based on the sentence structure.
Response 1: We agree with the reviewer's comment. We have changed the structure of the sentence. This point in revised manuscript and the detailed revision can be found in Line 185, Page5.
Point 2: Figure 4: To be consistent, the y-axis labels should read “initial day” instead of“initiate day”. Suggest correcting all figures with this axis.
Response 2: Thank you for pointing this out. We have replaced initial day” instead of “initiate day” in all figures of the paper.
Point 3: Line 428: “ …the initial day advanced by -0.93d/10a…” What does it mean to“advance” by a negative number? Maybe a different word would work better.
Response 3: We agree with the reviewer's comment.
We consulted the relevant articles. For the earlier start of growth season, usually used in advance, for the number should be changed to a positive number. the initial day advanced by 0.93d/10a. This point in revised manuscript and the detailed revision can be found in Line 526, Page17.
Point 4: Line 533: error “…in the region, It has certain…”, this should be a period, instead of a comma.
Response 4: Thank you for pointing this out. We have corrected it. This point in revised manuscript and the detailed revision can be found in Line 660, Page20.
Point 5: Line 535–536: Satellite data would serve as a nice supplement to this analysis.
Response 5: We agree with the reviewer's comment. Thank you very much. We have corrected it. This point in revised manuscript and the detailed revision can be found in Line 663, Page20.

Round 2
Reviewer 1 Report
I have no comments to the current version.